# Guidelines for Working Heights of the Lower-Limb Exoskeleton (CEX) Based on Ergonomic Evaluations

**DOI:** 10.3390/ijerph18105199

**Published:** 2021-05-13

**Authors:** Yong-Ku Kong, Chae-Won Park, Min-Uk Cho, Seoung-Yeon Kim, Min-Jung Kim, Dong Jin Hyun, Kihyeon Bae, Jong Kyu Choi, Sang Min Ko, Kyeong-Hee Choi

**Affiliations:** 1Department of Industrial Engineering, Sungkyunkwan University, Suwon 16419, Korea; ykong@skku.edu (Y.-K.K.); cwrachel@skku.edu (C.-W.P.); crayonmu@skku.edu (M.-U.C.); kimsy9035@skku.edu (S.-Y.K.); xlsk1013@skku.edu (M.-J.K.); 2Robotics Lab in the R&D Division, Hyundai Motor Company, Uiwang 16082, Korea; mecjin@hyundai.com (D.J.H.); kihyeon.bae@hyundai.com (K.B.); jongkyu.choi@hyundai.com (J.K.C.); sangminko@hyundai.com (S.M.K.)

**Keywords:** lower-limb exoskeleton, CEX, working height, harvesting task, work-related musculoskeletal disorders

## Abstract

The aim of this study was to evaluate the muscle activities and subjective discomfort according to the heights of tasks and the lower-limb exoskeleton CEX (Chairless EXoskeleton), which is a chair-type passive exoskeleton. Twenty healthy subjects (thirteen males and seven females) participated in this experiment. The independent variables were wearing of the exoskeleton (w/ CEX, w/o CEX), working height (6 levels: 40, 60, 80, 100, 120, and 140 cm), and muscle type (8 levels: upper trapezius (UT), erector spinae (ES), middle deltoid (MD), triceps brachii (TB), biceps brachii (BB), biceps femoris (BF), rectus femoris (RF), and tibialis anterior (TA)). The dependent variables were EMG activity (% MVC) and subjective discomfort rating. When wearing the CEX, the UT, ES, RF, and TA showed lower muscle activities at low working heights (40–80 cm) than not wearing the CEX, whereas those muscles showed higher muscle activities at high working heights (100–140 cm). Use of the CEX had a positive effect on subjective discomfort rating at lower working heights. Generally, lower discomfort was reported at working heights below 100 cm when using the CEX. At working heights of 100–140 cm, the muscle activity when wearing the CEX tended to be greater than when not wearing it. Thus, considering the results of this study, the use of the lower-limb exoskeleton (CEX) at a working height of 40–100 cm might reduce the muscle activity and discomfort of whole body and decrease the risk of related disorders.

## 1. Introduction

Work-related musculoskeletal disorders (WMSDs) of both the upper and lower extremities have become a major concern in many industries, such as the manufacturing, agricultural, forestry, and fishery sectors. WMSDs cause muscles, tendons, and nerves to be stressed and loaded due to excessive or repetitive exertive force, awkward body posture, less resting time, cold working environment, vibration, and so on. According to the Bureau of Labor Statistics [1], shoulder, back, and knee injuries comprise 15.2, 8.8, and 39.0%, respectively, of all WMSDs. There is evidence that a manual material handling (MMH) with the arms above the head, lifting weights, prolonged kneeling, and squatting can increase the risk of shoulder, back, and knee-related disorders [2,3].

The risk of musculoskeletal disorders in the agricultural sector is higher than in other sectors [4,5]. Awkward working postures occur for more than 50% of daily working hours during agricultural work [6,7]. Squatting and knee-bending, which often occur during agricultural work, cause pain in the back, legs, and knees. Such awkward postures during agricultural work, in particular, increase the rate of development of lower extremity musculoskeletal disorders among farmers [8]. Musculoskeletal disorders in the lower extremities are associated with tasks that require squatting or kneeling on the floor due to low crop height. In particular, harvesting at a low working height often causes pain in the lower extremities. Harvesting of peppers and eggplants are carried out at 20–40 cm from the ground, with the angle of the knee joint bent by more than 130°. The load at the knee joint when the knee is bent is 7.8 times greater than that experienced when the worker is standing [9]. When growing lettuce or strawberries, which exerts a high body burden, guidelines for appropriate heights to minimize body loads are needed because there are no clear standards for bed height [10,11,12]. The harvesting work is the most intense in grape fields, and the work is performed in a standing position for a long time at a working height of about 90–140 cm [13,14].

Many upper- and lower-limb wearable exoskeletons, which are mechanical structures worn on the body to enhance the power of the wearer, have been developed and studied to reduce exposure to prolonged stationary standing and sitting tasks [15,16]. Wearable exoskeletons are actively used in various fields, such as in military and industrial settings, as components of rehabilitation assistance, and on automobile assembly lines because the exoskeletons do not require separate spaces [17]. Usability evaluation studies have been conducted using EMG [4,18,19,20,21,22,23,24,25,26,27], body posture angle [26], heart rate [28], subjective discomfort rating [20,21,23,24,25,29], and performance [24,29] at actual industrial sites or in various laboratory environments.

In a laboratory environment, Bosch et al. [21] reported that subjective discomfort and muscle activities of the lumbar muscle (ES), trapezius (TR), and biceps femoris (BF) were reduced by about 20–44% during a simulated assembly task and static holding task when the participants wore an upper-limb Laevo exoskeleton. Heydari et al. [4] also reported a 17–23% reduction in activation in the lower back muscles (ES, LD: latissimus dorsi; EO: external oblique; IO: internal oblique), but no change in the abdominal rectus abdominis (RA) muscles, during a static holding task when participants used a personal lifting assistive device (PLAD) and wearable assistive device (WAD). Another static holding task study of a passive arm exoskeleton showed decreases in the activities of the biceps brachii (BB), medial deltoid (MD), erector spinae (ES), rectus abdominis (RA), biceps femoris (BF), rectus femoris (RF), tibialis anterior (TA), and gastrocnemius (GAS) muscles when participants wore the exoskeleton, which corresponded to a decrease in local discomfort [23].

Use of the Upper-limb Shoulder X [22] device during an overhead drilling task produced 34% and 18% reductions in the activities of the deltoid and the trapezius muscles, respectively. In contrast, the back muscle (ES) was not significantly affected by use of the exoskeleton. In a WAD study [20], the activities of the biceps and triceps were reduced by 40% and 56%, respectively, and subjective discomfort improved by 50% and 25% in the upper arm and shoulder, respectively, in the group with the device.

Luger et al. [25] and Steinhilber et al. [16] investigated the effects of wearing the Chairless Chair passive lower-limb exoskeleton on muscle activity and perceived discomfort during screwing, cable-mounting, and clip-fitting tasks at varying working distances. The results showed that the activities of the ES (erector spinae) and GAS (gastrocnemius) muscles were significantly lower during high siting and low sitting EXO tasks, respectively, than during standing (without EXO). The activities of the ES and GAS increased as the working distance increased. However, the activities of the trapezius and vastus lateralis muscles increased more during sitting with EXO than when the participants worked in standing postures.

Pillai et al. [27] tested the effects of wearing another passive lower-limb exoskeleton, the LegX, on the activities of the LES (lumbar erector spinae), TES (thoracic erector spinae), TA (tibialis anterior), RF (rectus femoris), ST (semitendinosus), and GAS (lateral gastrocnemius) muscles during drilling tasks at hip and knee height. They found a significant reduction in RF activity only (12–48%) when the exoskeleton was used.

Although it has been reported that the use of exoskeletons for agricultural work, as well as industrial work, can effectively reduce subjective discomfort and muscle fatigue [5], current exoskeleton-related studies consist mostly of upper limb studies (Lavitate AIRFRAME, Laevo exoskeleton, PLAD, WAD). Studies of lower-limb exoskeletons (Chairless chair, LegX) are rare and, in particular, few researchers have examined the applications of such exoskeletons in the agricultural field.

While some studies reported that use of an exoskeleton has a positive effect on all muscles [23,30,31,32], other studies have suggested that the load is simply transferred to other muscles. Use of some passive trunk exoskeletons, such as PLAD, HappyBack, and Bendezy, was associated with increased activity of the lower limb muscles [18,19,20,22]. Also, upper- and lower-limb exoskeleton studies are mainly focused on the upper and lower extremity muscles, respectively. Thus, it is necessary to examine muscle activities in the whole body with and without the exoskeleton.

Therefore, the purpose of this study was to evaluate the activities of upper and lower extremity muscles and subjective discomfort according to the heights of harvesting tasks and the use of a lower-limb exoskeleton (CEX).

## 2. Methods

### 2.1. Participants

This study was conducted on 20 healthy adults (thirteen males and seven females) who did not have any history of musculoskeletal pain or disorders. All participants were informed about the content and purpose of this study and provided a consent form before the start of the study. This research has been approved by the Sungkyunkwan University IRB (approval #: SKKU 2021-01-07). The anthropometric dimensions of the participants are shown in Table 1.

### 2.2. Apparatus

This CEX, a chair-type wearable exoskeleton developed by Hyundai Motors (Hyundai Motor Group, Seocho-gu, Seoul, Korea), was used to investigate the appropriate working height for industrial wearable robots during harvesting in this study (Figure 1). CEX, a type of passive exoskeleton, is a chair-type lower-limb exoskeleton that helps people work while performing seated tasks.

A passive exoskeleton provides increased physical strength using springs or dampers. It is more commonly used in workplaces than an active exoskeleton, which is primarily used for disabled people. The CEX is a relatively light model (1.6 kg vs. Ofress, 2.1 kg; Chairless Chair, 3.0 kg), and can be adjusted using Velcro straps to fit the worker’s body. Depending on the working height, the seating angle can also be set to three levels (55°, 70°, 85°). The CEX has three different sizes (small, medium, and large), and users can choose the appropriate size according to their height and weight. The small size is suitable for users who are 160 to 175 cm tall and weighing less than 80 kg. The medium and large sizes are suitable for 170~185 cm height (less than 100 kg weight) and 180~195 cm height (less than 120 kg weight), respectively. In addition, the length of calf and hip and sitting angle are also adjustable over 3–5 levels to body sizes and working conditions.

### 2.3. Experimental Procedure

Before the experiment, each participant was informed of relevant information, such as the purposes and procedures of this study and how to wear the exoskeleton, and the participant’s body dimensions were measured. In order to assess muscle activity in the upper and lower body during harvesting, the experiments were conducted after attachment of electromyography (EMG) equipment. A total of 16 muscles (eight upper-body muscles, two back muscles, and six lower-body muscles, each on the left and right sides of the body) were tested in this study.

To replicate a harvesting task similar to that seen in the agricultural field, a Styrofoam ball was attached to the laboratory wall with Velcro tape. The ball was attached at six different levels, ranging from 40 to 140 cm, at intervals of 20 cm; these distances were chosen considering the height of the harvesting work that causes physical stress and the height that can be reached when wearing the lower extremity exoskeleton (Figure 2).

Most previous research on lower-limb exoskeleton systems [13,25,33] was conducted at work heights of 100–140 cm, considering the reach of the arms, etc. Additionally, the typical work height, including the height of hydroponic cultivation, is reported to be 40 to 140 cm [9,10,11,12,13,14]; this presents a burden on the musculoskeletal system during harvesting of red pepper, lettuce, strawberry, and grape crops. Thus, in this study, the working heights were set to 40 to 140 cm.

After donning the CEX, all participants performed sufficient pre-work for 15 min to become familiar with the exoskeleton and the experimental task. Each participant also chose the most appropriate seated angle (55°, 70°, or 85°) for each working height Figure 3. At each height, all harvesting tasks were performed in a random order for two minutes each. A survey was conducted on physical discomfort and difficulty of work after the end of the task. After completing all 12 tasks, the participant rated the overall subjective discomfort rating of the CEX.

### 2.4. Experimental Design

Analysis of variance (ANOVA) was used to evaluate the effects of exoskeleton use, working height, and muscle type on muscle activity (electromyography: EMG) and subjective rating. Statistical analysis was performed using SPSS 20 (SPSS Inc, Chicago, IL, USA). Analysis of variance (ANOVA) and paired *t*-test were conducted to determine the effect of the CEX and working height on EMG activities and subjective rating. For multiple comparisons, Tukey’s test was also applied. The significance level of all tests was set at *p* < 0.05.

The independent variables were the use of the exoskeleton (w/ CEX, w/o CEX), working height (6 levels: 40, 60, 80, 100, 120, and 140 cm), and muscle type (8 levels: UT, ES, MD, TB, BB, BF, RF, and TA). All 12 trials (6 working heights × 2 levels of exoskeleton wearing) for each gender were completely randomized. The dependent variables were EMG activity (% MVC) and subjective rating, the definitions of which are as follows.

#### 2.4.1. EMG Activity (% MVC)

The TeleMyo 2400 DTS System (Noraxon, AZ, USA) was used to evaluate the effects of independent variables on all 16 muscle groups examined in this study. The upper trapezius (UT), erector spinae (ES), middle deltoid (MD), triceps brachii (TB), biceps brachii (BB), biceps femoris (BF), rectus femoris (RF), and tibialis anterior (TA) muscles on the left and right side of the body were chosen based on previous studies showing significant changes in the activity of these muscles with use of an exoskeleton [21,23,24,26,27]. Surface electrodes were positioned over the bellies of those muscles, parallel to the longitudinal axis of the muscle fibers, as recommended by Zipp [34] (Figure 4).

Each participant exerted two maximum voluntary contractions (MVCs) for 5 s for each of the 16 muscle groups. The average value of the 3 mid-trial seconds was used to normalize the muscle-specific activity. To minimize differences between participants and muscle groups, task EMG was normalized with regard to the MVC, based on the traditional formula shown in Equation (1) [35]. The EMG amplitudes for all muscle groups for each two-minute task were normalized and averaged in this study. The EMG amplitudes for resting were also collected.
Normalized EMG = [(Task EMG − resting EMG)/(Max. EMG − resting EMG)] × 100(1)

The EMG activity of muscles was acquired at a sampling frequency of 1500 Hz. The raw signals were digitally filtered using a 6th order Butterworth filter with a band pass filter (20–400 Hz), and then they were expressed as the root mean squared (RMS, 50 ms) signal [36]. Electrocardiogram (EGC) spikes can interfere with the EMG signals of muscles, especially when performed on the upper trunk and shoulder activities. Thus, an ECG reduction filter was applied to eliminate EGC signals in this study.

#### 2.4.2. Subjective Rating

At the end of each task, all participants were asked to provide subjective ratings of overall discomfort using the Borg RPE scale [37,38].

## 3. Results

### 3.1. EMG Amplitude (% MVC)

The main effects of working height and muscle type on the EMG amplitude were found to be statistically significant (all *p* < 0.001) (Figure 5).

The highest average muscle activities, 9.38 and 8.96% MVCs, were seen at the lowest (40 cm) and highest (140 cm) working height, respectively, and relatively low muscle activities (6.52–6.9% MVCs) were seen at the intermediate working height of 100–120 cm (Figure 5, left). The analysis of muscle activity by muscle type showed relatively high activity in the TB, ES, and UT, but less activity in the BB, RF, and TA (Figure 5, right). The main effect of exoskeleton use (w/ and w/o) on muscle activity (7.84 and 8.24% MVCs) was not statistically significant.

The interaction effect between the use of the exoskeleton and working height was statistically significant. Generally, similar levels of muscle activity (6.81–8.69% MVCs) were observed at working heights of 40–120 cm when wearing the CEX. On the other hand, without wearing the CEX (w/o), muscle activity differed according to working height. The highest muscle activity was (11.44% MVC) was observed at the lowest working height of 40 cm, which required a lot of bending of the knees and back, whereas muscle activity gradually decreased at heights over 100 cm (100 cm, 5.49% MVC; 120 cm, 5.12% MVC; and 140 cm, 6.02% MVC).

Interestingly, as shown in Figure 6, the muscle activities intersect depending on whether or not the participant wore the exoskeleton, starting between working heights of 80 and 100 cm. A task performed at a working height of 40–80 cm with the CEX (w/ CEX) tended to be associated with lower muscle activity than a task performed without the exoskeleton (w/o CEX). However, a task performed at a working height of 100–140 cm with the CEX (w/ CEX) was associated with higher muscle activity than one performed without the exoskeleton (w/o CEX).

The interaction effect between the use of the exoskeleton and muscle type was also statistically significant (*p* < 0.001). Overall, muscles in the upper extremities (UT, ES, MD, TB) showed higher activities (9.0 to 15.6% MVCs) when the participant wore the lower-limb exoskeleton. In contrast, the lower extremities (BF, RF, TA) showed significantly lower muscle activities (3.4 to 4.5% MVCs) when the participant did wear the exoskeleton. Specifically, the activity (4.5% MVC) of the BF was significantly lower with the exoskeleton than without it (9.3% MVC) (Figure 7).

The interaction effects between use of the exoskeleton, working height, and muscle type were also statistically significant (*p* < 0.001). The activity of the UT was the highest at working heights of 40–60 cm, which involved a good deal of knee flexion, when participants did not wear the CEX (w/o CEX), whereas the UT showed significantly higher muscle activity at 140 cm working height, which required a high shoulder flexion angle, when participants wore the CEX (w/ CEX) (Figure 8, left).

The ES showed the highest muscle activity at 80 cm working height, with a large back flexion angle, and the lowest muscle activity at 140 cm working height, similar to the normal standing posture, when the participant did not wear the exoskeleton (w/o CEX). The activity of the ES increased along with the working height when the participant wore the exoskeleton (w/ CEX) (Figure 8, right).

The lower extremity muscles showed significantly different trends depending on the working height, only when the participants did not wear the exoskeleton (w/o CEX). At a working height of 80 cm, with the most severe back flexion angle, the BF muscle showed the most considerable muscle activity. At the lower working heights of 40 and 60 cm (w/o CEX), the RF and TA showed high muscle activities, 12.6–15.7% and 6.4–8.5% MVCs, respectively. On the other hand, low muscle activities (2.3–6.1% MVCs) were seen at most working heights when the participants wore the exoskeleton (w/ CEX) (Figure 9).

### 3.2. Subjective Ratings

The main effects of working height and use of the exoskeleton on the subjective discomfort rating were statistically significant (*p* < 0.05). Overall, the subjective ratings (9.7–12.2 and 10.2) at lower working heights (40–80 cm) and without the exoskeleton (w/o CEX) were significantly greater than those (8.5–8.9 and 9.4) at higher working heights (100–140 cm) and with the exoskeleton (w/ CEX).

The interaction effect between working height and use of the exoskeleton on the subjective discomfort rating was also statistically significant (*p* = 0.001). When participants did not wear the exoskeleton (w/o CEX), the subjective discomfort ratings were significantly higher than when they did (w/ CEX) at lower working heights (less than 100 cm). Although the subjective discomfort ratings of the ‘w/ CEX’ group were slightly higher than those of the ‘w/o CEX’ group at working heights of over 100 cm, the difference was not statistically significant (Figure 10).

## 4. Discussion

This study was conducted to identify the applicable work height of a lower-limb exoskeleton for a harvesting task based on biomechanical data and subjective discomfort analysis.

Regardless of whether or not the lower-limb exoskeleton (CEX) is worn, the average muscle activities (9.38 and 8.96% MVCs) were greatest at the lowest working height (40 cm) and the highest working height (140 cm), which generally require high degrees of flexion of the knee and back and higher shoulder flexion angles, respectively. In particular, at lower working heights, the UT (13.7% MVC), which acts to elevate the scapula and provide upward/outwards rotation, RF (7.7% MVC), which performs pelvic and knee movements and plays a role in continuous posture maintenance, and TA muscles (9.4% MVC), which are necessary for ankle movements and balance, showed relatively higher muscle activities in the harvesting task. On the other hand, at higher working heights, the activities of the upper extremity muscle UT (14.5% MVC) and the MD (8.9% MVC) and TB (18.9% MVC) muscles, which mainly serve to move the shoulder and elbow, were significantly higher.

At lower working height (40–80 cm), which requires a high degree of knee and back flexion, most muscles (UT, ES, RF, and TA) showed lower muscle activity when wearing the CEX than that of when not wearing the CEX. The activities of the lower body muscles (BF, RF, and TA) were significantly reduced by use of the CEX, by 30.59–84.08%.

Although the CEX does not directly support the upper extremity muscles, the activity of upper body muscles (ES and UT muscle) decreased by 5.83–50.34% when the CEX was used. The UT muscle showed significantly reduced muscle activities when wearing the CEX by 35.93% (w/o CEX: 16.7% and w/ CEX: 10.7%) and 50.34% (w/o CEX: 14.7% and w/ CEX: 7.3%) at working heights of 40 and 60 cm, respectively. Similarly, the ES muscle also showed reduced muscle activity (5.83–29.48%) when wearing the CEX at a working height of 40–80 cm (w/o CEX: 12.0–17.3% and w/ CEX: 10.4–12.2%). These results are similar results to those of previous studies. Pillai et al. [27] reported a 12–48% reduction in RF muscle activity in a drilling task with use of a lower-limb exoskeleton. Other upper-limb exoskeleton studies also showed a 49–62% [23], 36–56% [20], 17–23% [4], and 20–44% [21] reduction in upper body muscle activities with use of the upper-limb exoskeleton.

According to our analysis, at the working height of 100–140 cm, which requires flexion of the shoulder rather than knee or back flexion, muscle activity tended to be higher when wearing the CEX than when not wearing it. This effect was particularly pronounced in the ES and UT upper extremity muscles, and increased significantly as the working height increased. In particular, at working heights of 100–140 cm, the UT muscle showed about 2.3–3.0 times greater muscle activity (1.9–22.6%) when working with CEX than when working without CEX (2.2–6.4%). TA muscles also significantly increased muscle activity when wearing the CEX than when not wearing the device at higher working heights (100, 120, 140 cm). However, the effect in these other muscles was not as clear as in the ES and UT.

According to the results of this study, use of the CEX can reduce the loads of both upper and lower body muscles at working heights under 100 cm. In contrast, the loads of upper body muscles might increase with use of the CEX at working heights over 100 cm.

The positive effect on subjective discomfort rating seen with use of the CEX at low working heights tended to be very similar to the analysis of muscle activity. Generally, lower discomfort ratings were seen at working heights below 100 cm when the CEX was used. In particular, at lower working heights (40–80 cm), the overall discomfort rating of using the CEX was significantly more positive than that of not using the CEX. Although the effect is not statistically significant, working heights of 120 and 140 cm tended to be associated with slightly higher overall discomfort when wearing the device.

Thus, considering the results of both muscle activity and subjective discomfort rating analysis in this study, wearing the CEX at a working height of 40–100 cm reduces the muscular loads and discomfort on the upper and lower bodies from a knee bending task. This could lead to a reduction in the risk of related disorders.

As mentioned in the Introduction, use of a lower-limb exoskeleton may be more effective at reducing the physical load and subjective discomfort of workers who are harvesting red peppers and eggplants (20–40 cm), lettuce (60–70 cm), strawberries (80–100 cm), and grapes (90 cm). On the other hand, harvesting of crops at heights above 120 cm, such as other types of strawberries (120–140 cm) and grapes (140 cm), is associated with less muscle load on the upper extremities when the lower limb exoskeleton is not worn.

The results of this study could be used to produce guidelines for the application of lower-limb exoskeletons to various working environments, including automobile assembly lines, as well as harvesting work. The current study used a relatively short and static task to simulate an actual harvesting task in a laboratory environment. Expansion of the findings to tasks with longer durations and more practical tasks will provide more valuable information in this research area.

This study has some limitations. First, this study was conducted only with young adults. Further study will recruit participants of various age groups and investigate the difference between the age groups. The second limitation is that this study did not consider the workload for a long-term period. In a further study, the tendency of workload over time would be considered by increasing the task time. Although this study has some limitations, the authors wish that the results of this study could be useful for reducing the prevalence rate of WMSDs in workers.

## 5. Conclusions

This study analyzed the muscle activity and subjective discomfort associated with use of the CEX based on working height. The results suggest that the utility of the CEX depends on the working height. In conclusion, it is recommended that workers use the lower extremity exoskeleton at working heights of 40 to 100 cm (below 100 cm).

This study may be applicable to working environments at various industrial sites, not just agricultural fields. Furthermore, more research into the use of exoskeletons should be done that is more active and applied on-site; such research could provide the underlying data for various usability evaluation studies that can reduce the physical stress of workers.

## Figures and Tables

**Figure 1 ijerph-18-05199-f001:**
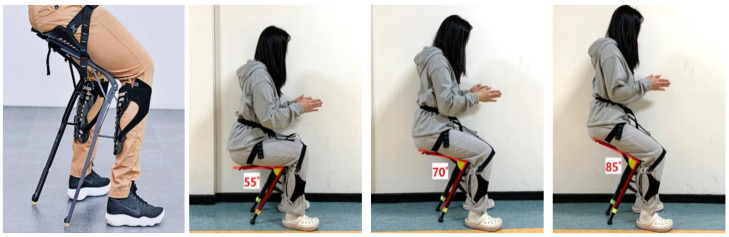
The CEX wearable lower-limb exoskeleton.

**Figure 2 ijerph-18-05199-f002:**
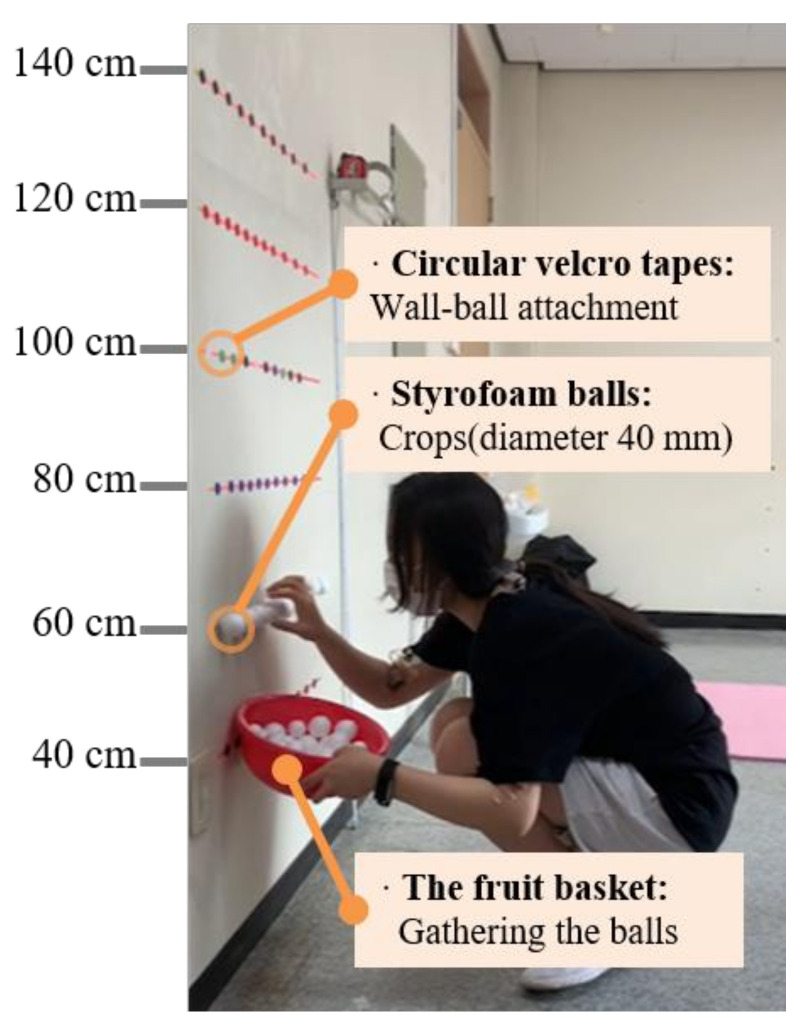
Experimental set-up with different working heights.

**Figure 3 ijerph-18-05199-f003:**
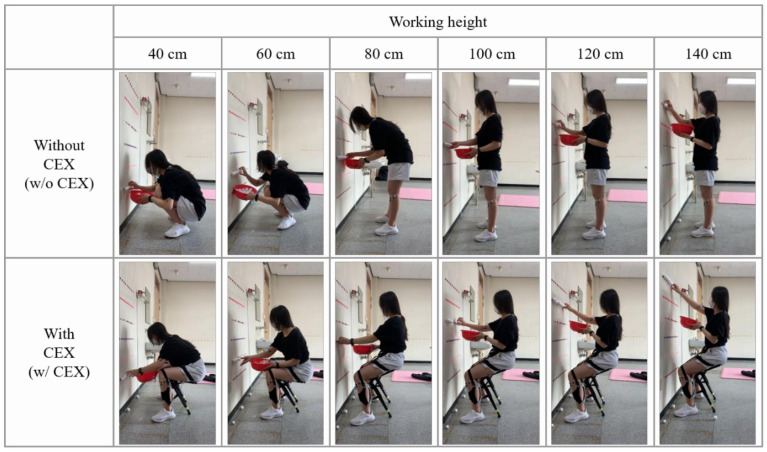
Experimental tasks according to working heights (w/o CEX (top) vs. w/ CEX (bottom)).

**Figure 4 ijerph-18-05199-f004:**
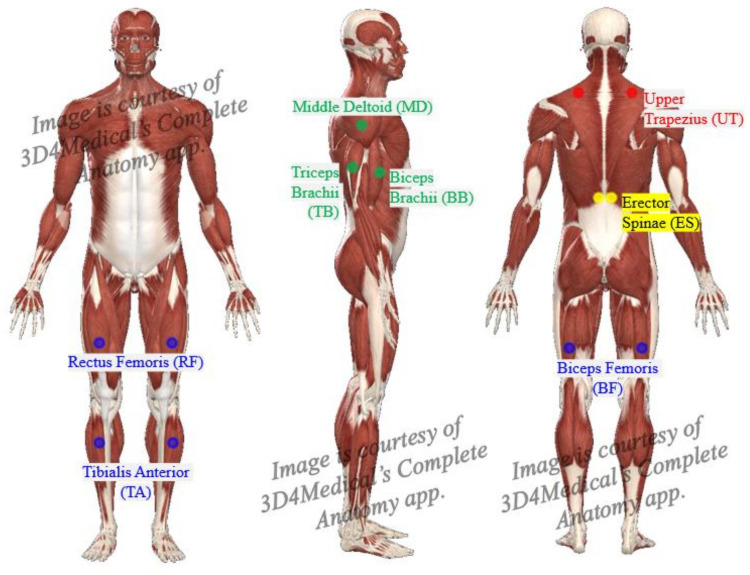
EMG sensor positions for all muscle groups.

**Figure 5 ijerph-18-05199-f005:**
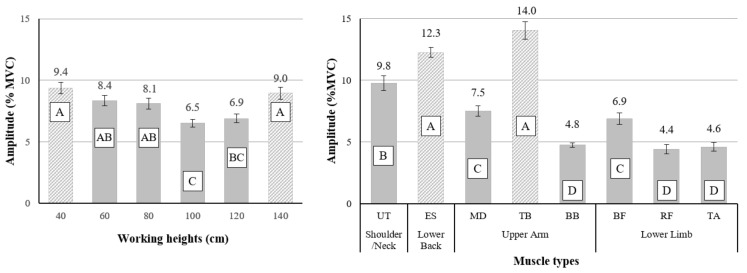
Main effects of working height (left) and muscle type (right). Different letters indicate significant statistical differences (*p* < 0.05, Tukey’s test).

**Figure 6 ijerph-18-05199-f006:**
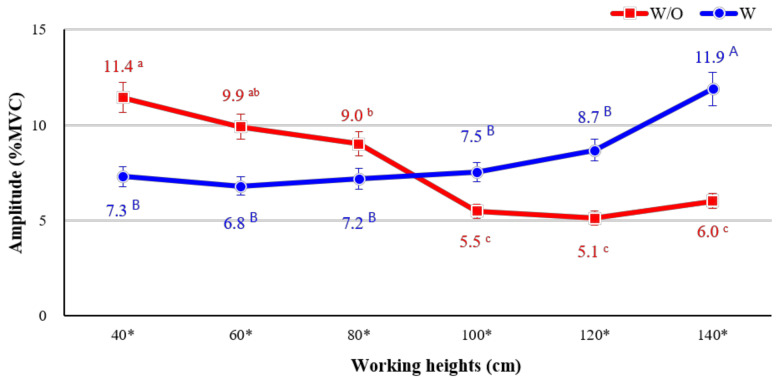
Interaction effect between use of the exoskeleton and working height. Asterisk (*) shows a significant difference between w/o and w/ (*p* < 0.05, Paired *t*-test). Different letters indicate significant statistical differences (*p* < 0.05, Tukey’s test).

**Figure 7 ijerph-18-05199-f007:**
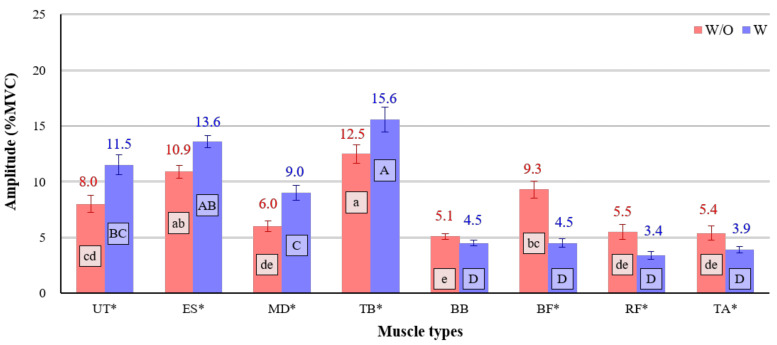
Interaction effect between use of the exoskeleton and muscle type. Asterisk (*) shows a significant difference between w/o and w/ (*p* < 0.05, Paired *t*-test). Different letters indicate significant statistical differences (*p* < 0.05, Tukey’s test).

**Figure 8 ijerph-18-05199-f008:**
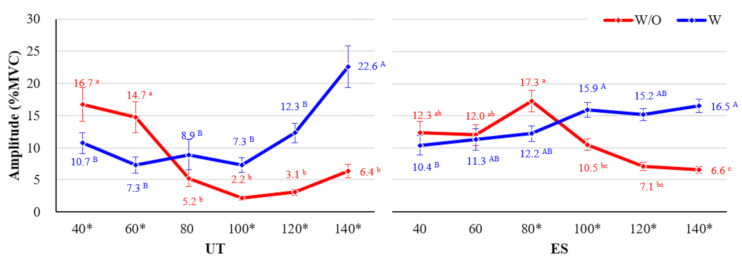
Interaction effects among wearing exoskeleton, working height, and muscle type (UT and ES). Asterisk (*) shows a significant difference between w/o and w/ (*p* < 0.05, Paired *t*-test). Different letters indicate significant statistical differences (*p* < 0.05, Tukey’s test).

**Figure 9 ijerph-18-05199-f009:**
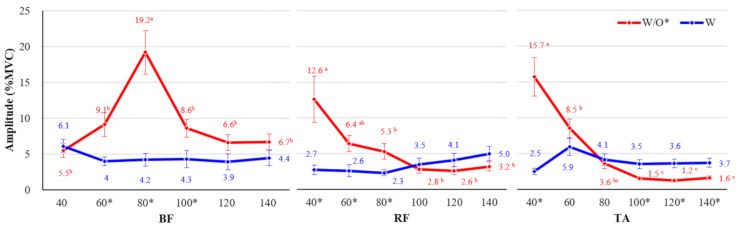
Interaction effects among wearing exoskeleton, working height, and muscle type (BF, RF, and TA). Asterisk (*) shows a significant difference between w/o and w/ (*p* < 0.05, Paired *t*-test). Different letters indicate significant statistical differences (*p* < 0.05, Tukey’s test).

**Figure 10 ijerph-18-05199-f010:**
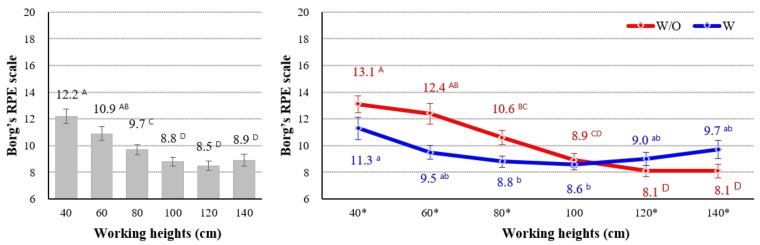
Interaction effect between exoskeleton use and working height. Asterisk (*) show significant difference between w/o and w/ (*p* < 0.05, Paired *t*-test). Different letters indicate significant statistical differences (*p* < 0.05, Tukey’s test).

**Table 1 ijerph-18-05199-t001:** Participants’ anthropometric data (mean ± standard deviation).

Gender	Age (Year)	Height (cm)	Weight (kg)
Male	22.5 ± 2.0	176.4 ± 4.2	72.0 ± 11.6
Female	20.7 ± 1.2	165.5 ± 4.4	57.2 ± 3.8

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
