# Peer review of "Guidelines for Working Heights of the Lower-Limb Exoskeleton (CEX) Based on Ergonomic Evaluations"

_ijerph, 2021, doi:10.3390/ijerph18105199_

Round 1

Reviewer 1 Report

General comments:

The objective of this study is to assess the influence of wearing a chairless exoskeleton on muscle activity while performing a picking tasks at different heights. To this aim, the authors collected EMG data of 16 muscles during a simulated harvesting task at 6 different heights (40 to 140cm) with and without the wearable sitting support CEX. The introduction is well written however the objectives of the study have to be clarified, as the secondary objective is a consequence of the first one. The methodology is well explained even if some details are missing (see below). The Results section tackled some questions although there have not been introduced, e.g. influence of gender and body parts. Thus the definition of independent variables has to be clarified.  The use of an average amount of EMG (across all muscles) is very questionable as it does not look biomechanically consistent. To correspond to the aim of the study, main results should be dedicated to the interaction effects between use of the exoskeleton, working height, and muscle activity. Standard deviations are missing and the design of graphics can also be questioned. In the Discussion section, the first paragraph about the influence of gender appears awkward and unnecessary. Some outcomes in the Discussion are difficult to find in the Results section, like the fact that wearing the exoskeleton would decrease EMG activity by 30%. Consequently, for all these reasons, a major revision is recommended.

Specific comments:

Abstract

The distribution of “participants” has to be detailed in the abstract: number of women and men.

The fact that “gender” is an independent variable is questionable as it does not correspond to the main objective of the current study. Removing this question might help clarify the paper.

The terms “body part” might be detailed as it refers to muscle activity on different anatomical areas.

“the average muscle activity of participants wearing the CEX was approximately 30% lower than that of participants without the CEX”: this sentence is awkward as it gives the impression that two groups of participants did the experiment, please clarify. Moreover, the number “30% lower” is difficult to find into the Results section (see below).

…” at a working height of 40-100cm might reduce the muscular load and discomfort of muscles and decrease the risk of related disorders.”: this part of the sentence has to be reviewed as it deals with “muscle activity” and not “muscle load”, and also “global discomfort” and not “discomfort of muscles”.

Introduction

  • Page 2, l.58: “…to reduce exposure to prolonged stationary standing and sitting tasks”. The term “prolonged” is an important point here: wearable exoskeletons can be useful on a long-term task however the temporal aspect has not been studied in the current study, please justify.
  • Page 3, l. 112: “… and (2) to provide guidelines for the appropriate working height for use of the lower-limb exoskeleton.”. This is not an objective but an application of the current study, please remove.

Methods

  • Page 3, l.122: “The elbow height…” Why has this distance been measured? It is not reused in the next sections and it does not support hypotheses of the study. Please clarify.
  • Page 3, l. 134: “…can be adjusted using Velcro straps to fit the worker’s body.”. Is there any anthropometrical requirement to wear the CEX (minimal and maximal stature, etc.)? Please detail.
  • Page 4, l. 162: “After donning the CEX, all participants performed sufficient pre-work to become familiar with the exoskeleton and the experimental task.”. How long was this familiarizing phase?
  • Page 5, l. 179: the “gender” independent variable might be removed to improve the reading of the article. The terms “body parts” need to be replaced by “muscle activity” in the whole paper.
  • Page 6, l. 202: “… based on the traditional formula shown in equation (1).”. What is the reference related to this formula? The resting EMG is usually not removed and the maximal EMG (after a maximal voluntary contraction or the maximal value during the task) represents 100%. Current EMG is then expressed as a proportion of this maximum for normalization. Please justify.

Results

  • As previously mentioned, results concerning the effects of gender might be removed to clarify the paper.
  • An average EMG activity (for all muscles) is often used to express results in this study. However, this average amount is very questionable as it is not biomechanically consistent (none of the reference article in biomechanics used this average data). Indeed, each muscle has its own characteristics so averaging EMG through different muscles would remove this specificity. An EMG-based analysis should be performed per muscle only. The Results section should be thus reviewed consequently.
  • Standard deviations (SD) are missing in all results, both in the text and graphics, please add.
  • Figure 5: what are letters (A, B, C, D, AB, BC) referring to? Please detail in the caption.
  • Figure 6: please remove as it refers to the question of gender.
  • Figure 7: what are letters (a, b, A, AB, etc.) referring to? Please detail in the caption.
  • Figure 8: why using lines between muscle EMG data? There is no rationale for that, please use histograms. SD are missing and significance should be integrated into the figure.
  • Figure 9: what are letters referring to? What is the aim of this figure? Showing that EMG activity is different with heights, wearing or not an exoskeleton? This is obvious and does not correspond to any question from the literature. Thus I would propose to remove the interaction between working heights and muscle activity from the analysis as it undermines the main message of the paper: what is the impact of a lower-limb exoskeleton during harvesting task at different heights?
  • Figure 10 and 11: explanation about letters, SD and significance are necessary.
  • Page 10, l.313: “The interaction effect between working height and use…” it is necessary to precise that it concerns subjective ratings in this sentence.
  • Figure 12: the two graphs need to be described in the caption. Explanation about letters, SD and significance are necessary.

Discussion

  • The first paragraph about the influence of gender is awkward and should be removed.
  • Page 11, l.336: “In particular, at lower working heights, …”. This sentence is too long and should be segmented.
  • Page 11, l.346: “… was approximately 30% lower…” I don’t find where this number comes from. Is based on average EMG? If yes, it has to be reviewed. This kind of sentence might take place if all muscles present a decrease of 30% of activity in all conditions with the exoskeleton, which is not the case.
  • Page 11, l.360: “… also showed slightly…” the word “slightly” needs to be removed as it is not scientifically consistent, please be more accurate (significant or not?).

Miscellaneous

  • Page 3, l. 100: “…(Chirless chair, LegX)…” the word “chirless” has to be replaced by the word “chairless”

Author Response

# Reviewer 1

General comments:

The objective of this study is to assess the influence of wearing a chairless exoskeleton on muscle activity while performing a picking tasks at different heights. To this aim, the authors collected EMG data of 16 muscles during a simulated harvesting task at 6 different heights (40 to 140cm) with and without the wearable sitting support CEX. The introduction is well written however the objectives of the study have to be clarified, as the secondary objective is a consequence of the first one [Comment #7]. The methodology is well explained even if some details are missing (see below). The Results section tackled some questions although there have not been introduced, e.g. influence of gender and body parts. Thus, the definition of independent variables has to be clarified [Comments #2; #3, #11, #13, #17, #24]. The use of an average amount of EMG (across all muscles) is very questionable as it does not look biomechanically consistent [Comments #4, & #14]. To correspond to the aim of the study, main results should be dedicated to the interaction effects between use of the exoskeleton, working height, and muscle activity. Standard deviations are missing and the design of graphics can also be questioned [Comment #15]. In the Discussion section, the first paragraph about the influence of gender appears awkward and unnecessary [Comments #24]. Some outcomes in the Discussion are difficult to find in the Results section, like the fact that wearing the exoskeleton would decrease EMG activity by 30% [Comment #4, #26]. Consequently, for all these reasons, a major revision is recommended.

: Thank you for your valuable comments. The authors tried to reflect the reviewer's comments and revised this manuscript as follows.

 Specific comments:

Abstract

[Comment #1]: The distribution of “participants” has to be detailed in the abstract: number of women and men.

: Thank you for your valuable comment. As your request, the detailed information about participants has been added as follows. 

[Page 1, Lines 15-16: “Twenty healthy subjects (thirteen males and seven females) participated in this experiment.”]

[Comment #2]: The fact that “gender” is an independent variable is questionable as it does not correspond to the main objective of the current study. Removing this question might help clarify the paper.

: The gender section has been deleted as the reviewer’s comment to help clarify this study. 

[Comment #3]: The terms “body part” might be detailed as it refers to muscle activity on different anatomical areas.

: The authors agree with the reviewer’s comment. To clarify which muscle groups of the body were active, the authors replaced ‘body part’ with ‘muscle type’ in the manuscript.

[Page 1, Lines 16-19: “The independent variables were wearing of the exoskeleton (w/ CEX, w/o CEX), working height (6 levels: 40, 60, 80, 100, 120, 140 cm), and muscle type (8 levels:  upper trapezius (UT), erector spinae (ES), middle deltoid (MD), triceps brachii (TB), biceps brachii (BB), biceps femoris (BF), rectus femoris (RF), and tibialis anterior (TA))”.]

[Comment #4]: “the average muscle activity of participants wearing the CEX was approximately 30% lower than that of participants without the CEX”: this sentence is awkward as it gives the impression that two groups of participants did the experiment, please clarify. Moreover, the number “30% lower” is difficult to find into the Results section (see below).

: At the height of 40cm, W CEX (7.3% MVC) showed 36.0% lower EMG average amplitude than that of W/O (11.4% MVC) in Figure 7. The average amplitude of 60 and 80cm height also showed a similar pattern and showed 31.3% and 20.0% lower amplitude than W/O, respectively. To clarify the muscle activities in w/ and w/o the CEX for different working heights, the authors tried to present them based on each muscle group in this manuscript. So, it has been revised as follows.  

[Page 1, Lines 20-22: “When wearing the CEX, UT, ES, RF, and TA showed lower muscle activities at low working heights (40–80 cm) than not wearing the CEX, whereas those muscles showed higher muscle activities at high working heights (100-140 cm).”]

[Comment #5]: “… at a working height of 40-100cm might reduce the muscular load and discomfort of muscles and decrease the risk of related disorders.”: this part of the sentence has to be reviewed as it deals with “muscle activity” and not “muscle load”, and also “global discomfort” and not “discomfort of muscles”.

: Thank you for your comment. ‘Muscle load’ and ‘discomfort of muscles’ have been modified to ‘muscle activity’ and ‘discomfort of whole body’, respectively.

[Page 1, Lines 27-28: “~ 40-100cm might reduce the muscle activity and discomfort of whole body and decrease the risk of related disorders.”]

  1. Introduction

[Comment #6]: Page 2, l.58: “…to reduce exposure to prolonged stationary standing and sitting tasks”. The term “prolonged” is an important point here: wearable exoskeletons can be useful on a long-term task however the temporal aspect has not been studied in the current study, please justify.

: Thank you for your valuable comment. The authors very much agree with the reviewer's opinion. To verify the effectiveness of wearing EXO in the workplace, we think the study of the temporal aspect will be very important. First of all, the purpose of the current study was to research the effect of wearing EXO on muscle activity and subjective discomfort in a relatively short period of time, and we would like to conduct a prolonged task study later on.

The limitations of this study were added in the Discussion section.

[Page 11, Lines 388-390: “The second limitation is that this study did not consider the workload for a long-term period. In a further study, the tendency of workload over time would be considered by increasing the task time.”]

[Comment #7]: Page 3, l. 112: “… and (2) to provide guidelines for the appropriate working height for use of the lower-limb exoskeleton.”. This is not an objective but an application of the current study, please remove.

: The second purpose has been removed as requested. 

[Page 3, Lines 112-114: “Therefore, the purpose of this study was to evaluate the activities of upper and lower extremity muscles and subjective discomfort according to the heights of harvesting tasks and the use of a lower-limb exoskeleton (CEX).”]

  1. Methods

[Comment #8]: Page 3, l.122: “The elbow height…” Why has this distance been measured? It is not reused in the next sections and it does not support hypotheses of the study. Please clarify.

: The authors agree with the reviewer’s opinion. Thus, the elbow height has been deleted in Table 1 and section 2.1.

[Comment #9]: Page 3, l. 134: “…can be adjusted using Velcro straps to fit the worker’s body.”. Is there any anthropometrical requirement to wear the CEX (minimal and maximal stature, etc.)? Please detail.

: Thanks for your comment. The anthropometric requirement of the CEX has been added in the 2.2 section.

[Page 3, Lines 136-141: “The CEX has three different sizes (small, medium, and large), and users can choose the appropriate size according to their height and weight. The small size is suitable for users who are 160 to 175cm tall and weighing less than 80kg. The medium and large sizes are suitable for 170~185cm height (less than 100kg weight) and 180~195cm height (less than 120kg weight), respectively. In addition, the length of calf and hip and sitting angle are also adjustable over 3-5 levels to body sizes and working conditions.”]

[Comment #10]: Page 4, l. 162: “After donning the CEX, all participants performed sufficient pre-work to become familiar with the exoskeleton and the experimental task.”. How long was this familiarizing phase?

: In this study, 15 minutes of pre-work time was provided to all participants. It has been added in section 2.3.

[Page 5, Lines 167-168: “After donning the CEX, all participants performed sufficient pre-work for 15 minutes to become familiar with the exoskeleton and the experimental task.”]

[Comment #11]: Page 5, l. 179: the “gender” independent variable might be removed to improve the reading of the article. The terms “body parts” need to be replaced by “muscle activity” in the whole paper.

: Gender has been deleted in section 2.4, and authors thought that it is more appropriate to replace ‘body part’ with ‘muscle type’.

[Page 5, Lines 186-188: “The independent variables were the use of the exoskeleton (w/ CEX, w/o CEX), working height (6 levels: 40, 60, 80, 100, 120, 140 cm), and muscle type (8 levels: upper trapezius (UT), erector spinae (ES), middle deltoid (MD), triceps brachii (TB), biceps brachii (BB), biceps femoris (BF), rectus femoris (RF), and tibialis anterior (TA).”]

[Comment #12]: Page 6, l. 202: “… based on the traditional formula shown in equation (1).”. What is the reference related to this formula? The resting EMG is usually not removed and the maximal EMG (after a maximal voluntary contraction or the maximal value during the task) represents 100%. Current EMG is then expressed as a proportion of this maximum for normalization. Please justify.

: Normalization of EMG data is a method to expressing the task muscle activity as some percentage of the reference value. In this study, we used the maximum voluntary contraction (MVC) value as a maximum reference. Quite often, an EMG resting value is also used in order to quantify the resting level activity, and this could become a low-end reference (Soderverg, 1991; Serroussi and Pope, 1987), and the normalized EMG value more accurately reflects the muscle’s activity level required to perform the task (Mirka, 1991). Mirka (1991) defined this normalization method, and the equation is as follows [1]. Reference has been added in Method and Reference section.

[Page 6, Lines 209-210: “~ based on the traditional formula shown in equation (1) [35].”]

[Page 13, Line 491: “35. Mirka, G.A. The quantification of EMG normalization error, Ergon. 1991, 34(3), 343-352.”].

  1. Results

[Comment #13]: As previously mentioned, results concerning the effects of gender might be removed to clarify the paper.

: All results about gender have been removed. Thank you for your valuable comments.

[Comment #14]: An average EMG activity (for all muscles) is often used to express results in this study. However, this average amount is very questionable as it is not biomechanically consistent (none of the reference article in biomechanics used this average data). Indeed, each muscle has its own characteristics so averaging EMG through different muscles would remove this specificity. An EMG-based analysis should be performed per muscle only. The Results section should be thus reviewed consequently.

: Thank you for your valuable comments. In Figure 5 (left) and 7 in the Result section, the average value of EMGs was inevitably used to examine the overall trend of muscle activities associated with working height and the interaction effect between working height and use of exoskeleton. Although the authors agree with the reviewer’s comment, these figures would be a useful way to understand the overall trend. However, as the comment of the reviewer (the authors also understand that the analysis of each muscle is important), the results of each muscle are presented through Figures 8, 10, and 11. 

As the reviewer’s comment, some of the contents where the average value was used have been removed, and muscle-specific analysis has been added in the Abstract and Discussion section.

[Page 1, Lines 20-22: “When wearing the CEX, UT, ES, RF, and TA showed lower muscle activities at low working heights (40–80 cm) than not wearing the CEX, whereas those muscles showed higher muscle activities at high working heights (100-140 cm).”]

[Page 10, Lines 335-337: “In lower working height (40-80cm), which requires a high degree of knee and back flexion, most muscles (UT, ES, RF, and TA) showed lower muscle activity when wearing the CEX than that of when not wearing the CEX.”]

[Page 10, Lines 341-345: “UT muscle showed significantly reduced muscle activities when wearing the CEX by 35.93% (w/o CEX: 16.7% and w/ CEX: 10.7%) and 50.34% (w/o CEX: 14.7% and w/ CEX: 7.3%) at working height of 40 and 60cm, respectively. Similarly, ES muscle also showed reduced muscle activity (5.83-29.48%) when wearing the CEX at a working height of 40-80cm (w/o CEX: 12.0-17.3% and w/ CEX: 10.4-12.2%)”]

[Pages 10, Lines 354-356: “In particular, at working heights of 100-140cm, the UT muscle showed about 2.3-3.0 times greater muscle activity (1.9-22.6%) when working with CEX than when working without CEX (2.2-6.4%).”]

[Comment #15]: Standard deviations (SD) are missing in all results, both in the text and graphics, please add.

: Based on the reviewer 1, 2, and 3, standard errors have been added in all figures.  

[Comment #16]: Figure 5: what are letters (A, B, C, D, AB, BC) referring to? Please detail in the caption.

: Thank you for your comment. The caption has been added in Figure 5 as follow.

[Page 7, Lines 238-239: “Different letters indicate significant statistical differences (p< 0.05, Tukey’s test).”]

[Comment #17]: Figure 6: please remove as it refers to the question of gender.

: Figure 6 and related contents have been removed based on the reviewer's request.

[Comment #18]: Figure 7: what are letters (a, b, A, AB, etc.) referring to? Please detail in the caption.

: The caption has been added in Figure 7.  

[Page 7, Lines 258-259: “Asterisk (*) shows a significant difference between W/O and W (p<0.05, Paired t-test). Different letters indicate significant statistical differences (p<0.05, Tukey’s test).”]

[Comment #19]: Figure 8: why using lines between muscle EMG data? There is no rationale for that, please use histograms. SD are missing and significance should be integrated into the figure.

: As the reviewer’s comments, Figure 8 has been revised.

[Page 8, Lines 270-272: “Asterisk (*) shows a significant difference between W/O and W (p<0.05, Paired t-test). Different letters indicate significant statistical differences (p<0.05, Tukey’s test).”]

Figure 7. Interaction effect between use of the exoskeleton and muscle type. Asterisk (*) shows a significant difference between W/O and W (p< 0.05, Paired t-test). Different letters indicate significant statistical differences (p< 0.05, Tukey’s test).

{Figure 8 has been changed to Figure 7 due to the deletion of Figure 6}

[Comment #20]: Figure 9: what are letters referring to? What is the aim of this figure? Showing that EMG activity is different with heights, wearing or not an exoskeleton? This is obvious and does not correspond to any question from the literature. Thus, I would propose to remove the interaction between working heights and muscle activity from the analysis as it undermines the main message of the paper.

: Thank you for your valuable comment. The authors agreed with the reviewer’s opinion. Figure 9 and contents have been removed in this manuscript.

[Comment #21]: Figure 10 and 11: explanation about letters, SD and significance are necessary.

: Figures 10 and 11 have been revised. The results of paired t-test (W/O vs. W) have also been added. The bar graphs have been replaced with line graphs to improve legibility.

[Page 8, Lines 287-289 & Page 9, Lines 298-300: “Asterisk (*) shows a significant difference between W/O and W (p< 0.05, Paired t-test). Different letters indicate significant statistical differences (p< 0.05, Tukey’s test).”]

Figure 8. Interaction effects among wearing exoskeleton, working height, and muscle types (UT and ES). Asterisk (*) shows a significant difference between W/O and W (p< 0.05, Paired t-test). Different letters indicate significant statistical differences (p< 0.05, Tukey’s test).

{Figure 10 has been changed to Figure 8 due to the deletion of Figures 6 and 9}

Figure 9. Interaction effects among wearing exoskeleton, working height, and muscle type (BF, RF, and TA). Asterisk (*) shows a significant difference between W/O and W (p< 0.05, Paired t-test). Different letters indicate significant statistical differences (p< 0.05, Tukey’s test).

{Figure 11 has been changed to Figure 9 due to the deletion of Figures 6 and 9}

[Comment #22]: Page 10, l.313: “The interaction effect between working height and use…” it is necessary to precise that it concerns subjective ratings in this sentence.

: The sentence has been revised as requested.

[Page 9, Line 308-3091: “The interaction effect between working height and use of the exoskeleton on the subjective discomfort rating was also statistically significant (p= 0.001).”]

[Comment #23]: Figure 12: the two graphs need to be described in the caption. Explanation about letters, SD and significance are necessary.

: Figures 12 has been revised.

[Page 9, Lines 317-3181: “Asterisk (*) shows a significant difference between W/O and W (p< 0.05, Paired t-test). Different letters indicate significant statistical differences (p< 0.05, Tukey’s test).”]

Figure 10. Interaction effect between exoskeleton use and working height. Asterisk (*) show significant difference between W/O and W (p<0.05, Paired t-test). Different letters indicate significant statistical differences (p<0.05, Tukey’s test).

{Figure 12 has been changed to Figure 10 due to the deletion of Figures 6 and 9}

  1. Discussion

[Comment #24]: The first paragraph about the influence of gender is awkward and should be removed.

: All contents related to the gender effect have been removed.

[Comment #25]: Page 11, l.336: “In particular, at lower working heights, …”. This sentence is too long and should be segmented.

: Thank you for your comment. That sentence has been condensed.

[Page 10, Lines 367-368: “In particular, at lower working heights (40-80 cm), the overall discomfort rating of using the CEX was significantly more positive than that of not using the CEX.”]

[Comment #26]: Page 11, l.346: “… was approximately 30% lower…” I don’t find where this number comes from. Is based on average EMG? If yes, it has to be reviewed. This kind of sentence might take place if all muscles present a decrease of 30% of activity in all conditions with the exoskeleton, which is not the case.

: Thank you for your comments. As recommendations, the average data of muscles have been replaced with muscle-specific results.  

[Page 10, Lines 336-338: “In lower working height (40-80cm), which requires a high degree of knee and back flexion, most muscles (UT, ES, RF, and TA) showed lower muscle activity when wearing the CEX than that of when not wearing the CEX.”]

[Page 10, Lines 342-346: “UT muscle showed significantly reduced muscle activities when wearing the CEX by 35.93% (w/o CEX: 16.7% and w/ CEX: 10.7%) and 50.34% (w/o CEX: 14.7% and w/ CEX: 7.3%) at working height of 40 and 60cm, respectively. Similarly, ES muscle also showed reduced muscle activity (5.83-29.48%) when wearing the CEX at a working height of 40-80cm (w/o CEX: 12.0-17.3% and w/ CEX: 10.4-12.2%)”]

[Pages 10, Lines 355-357: “In particular, at working heights of 100-140cm, the UT muscle showed about 2.3-3.0 times greater muscle activity (1.9-22.6%) when working with CEX than when working without CEX (2.2-6.4%).”]

[Comment #27]: Page 11, l.360: “… also showed slightly…” the word “slightly” needs to be removed as it is not scientifically consistent, please be more accurate (significant or not?).

: Thank you for your comments. As the reviewer’s comment, this sentence has been revised more clearly. For clarity of this study, the result of paired t-test between W and W/O CEX has been added in Figures 10 & 11.

[Page 10, Lines 357-359: “TA muscles also significantly increased muscle activity when wearing the CEX than when not wearing the device at higher working heights (100, 120, 140cm).”]

  1. Miscellaneous

[Comment #28]: Page 3, l. 100: “…(Chirless chair, LegX)…” the word “chirless” has to be replaced by the word “chairless” : Typo has been revised.

Reviewer 2 Report

This study used a wearable passive exoskeleton to compare EMG during various work tasks, and the results are interesting. However, there are some concerns to be clarified.   

  • Line 135, could the length or height of the exoskeleton be adjusted? If not, the protocol had a problem for different individuals.
  • Line 176, what specific functions were selected in Spss?
  • Line 204, how to determine or collect rest EMG signals, e.g. standing or siting?
  • Line 207 and 208, what is RMS and how to calculate it? What is ECG filters? Please give detailed introduction.
  • Figure 5, where are SD or SER?
  • In Figure 6 and other figures, what meanings of superscripts a,b,c,d..or A,B,C,D…etc? Also in Figure 6, with or without the wearable exoskeletal device?

Author Response

# Reviewer 2

This study used a wearable passive exoskeleton to compare EMG during various work tasks, and the results are interesting. However, there are some concerns to be clarified.   

: Thank you for your valuable comments. The authors tried to reflect the reviewer's comments and revised this manuscript as follows.

[Comment #1]: Line 135, could the length or height of the exoskeleton be adjusted? If not, the protocol had a problem for different individuals.

: Thank you for your valuable comments. The CEX has three types of sizes (S, M, and L), and users can choose the appropriate size according to their height and weight. In addition, the length of calf and hip and sitting angle of CEX can be adjusted in 3-5 steps. It has the advantage of being able to adjustable according to the user’s body. More information about the CEX has been described in this manuscript.

[Page 3, Lines 136-141: “The CEX has three different sizes (small, medium, and large), and users can choose the appropriate size according to their height and weight. The small size is suitable for users who are 160 to 175cm tall and weighing less than 80kg. The medium and large sizes are suitable for 170~185cm height (less than 100kg weight) and 180~195cm height (less than 120kg weight), respectively. In addition, the length of calf and hip and sitting angle are also adjustable over 3-5 levels to body sizes and working conditions.”]

[Comment #2]: Line 176, what specific functions were selected in Spss?

: Analysis of variance (ANOVA) and paired t-test were used in SPSS. Related information has been added in the method section.

[Page 5, Lines 182-183: “Analysis of variance (ANOVA) and paired t-test were conducted to determine the effect of the CEX and working height on EMG activities and subjective rating.”]

[Comment #3]: Line 204, how to determine or collect rest EMG signals, e.g. standing or siting?

: Thank you for your comments. To measure the resting EMG of the ES muscle, all participants were asked to lie on a pad in a prone position and asked to be relaxed for 5 seconds, and it was repeated two times. The resting EMGs of other muscles were also measured in sitting posture for two trials and used the average value of two trials in this study.

[Comment #4]: Line 207 and 208, what is RMS and how to calculate it? What is ECG filters? Please give detailed introduction.

: Thank you for your comments. First, the RMS represents the squared root of the average poser of the EMG signals for a given time. It is the recommended quantitative index of muscle activity for surface EMG signals (Basmajian and De Lura, 1985). Related information has been added in the method and reference sections.

Secondly, ECG (electrocardiogram) spikes can interfere with the EMG recording of muscles, especially when performed on the upper trunk & shoulder movements. Therefore, an ECG reduction filter has been widely used to eliminate ECG signals. Detailed information has been added in the method section.

[Page 6, Line 216: “~ then expressed as the root mean squared (RMS, 50 ms) signal [36].”]

[Page 6, Lines 217-219: “Electrocardiogram (EGC) spikes can interfere with the EMG signals of muscles, especially when performed on the upper trunk and shoulder activities. Thus, an ECG reduction filter was applied to eliminate EGC signals in this study.”]

[Comment #5]: Figure 5, where are SD or SER?

: Standard errors have been added in all figures.  

[Comment #6]: In Figure 6 and other figures, what meanings of superscripts a,b,c,d..or A,B,C,D…etc? Also in Figure 6, with or without the wearable exoskeletal device?

: Thank you for your valuable comments. Figure 6 has been deleted as reviewer 1’s opinion. However, as per your comment, the caption has been added to all figures. For example,

[Page 7, Lines 259-259: “Asterisk (*) shows a significant difference between W/O and W (p< 0.05, Paired t-test). Different letters indicate significant statistical differences (p< 0.05, Tukey’s test).”]

Reviewer 3 Report

The paper proposes guidelines for working heights of lower-limb exoskeletons in relation to ergonomic parameters. As exoskeletons are an important area of current research this topic of of great scientific interest. As the authors state in the introduction, there is a distinct lack of ergonomic recommendations for the lower limb (e. g. for work heights when preforming different tasks), in contrast to upper limb exoskeletons. Thus, the study contributes to scientific progress.

The study design is executed with sound scientific methods and data and methods are described in a clearly structured manner. The technical description of the exo device is comprehensive and the tasks are clearly defined and standardized. Using EMG and Borg-Scale for subjective comfort are sound scientific means and the statistical evaluation was cared out appropriately. Results are displayed with appropriate graphics and sufficiently described verbally. The discussion relates summarizes the actual results correctly and puts it in matching relation to related work. Overall, the paper is well-structured and written in a clear and concise manner.

In the following there is a list of two minor issues that should be addressed before publication:

  • The test person collective is quite young. You should mention this potential bias.

  • I suggest adding error bars in Figs. 6-8 to visualize the standard deviation inside the data.

Author Response

# Reviewer 3

The paper proposes guidelines for working heights of lower-limb exoskeletons in relation to ergonomic parameters. As exoskeletons are an important area of current research this topic of great scientific interest. As the authors state in the introduction, there is a distinct lack of ergonomic recommendations for the lower limb (e. g. for work heights when preforming different tasks), in contrast to upper limb exoskeletons. Thus, the study contributes to scientific progress.

The study design is executed with sound scientific methods and data and methods are described in a clearly structured manner. The technical description of the exo device is comprehensive and the tasks are clearly defined and standardized. Using EMG and Borg-Scale for subjective comfort are sound scientific means and the statistical evaluation was cared out appropriately. Results are displayed with appropriate graphics and sufficiently described verbally. The discussion relates summarizes the actual results correctly and puts it in matching relation to related work. Overall, the paper is well-structured and written in a clear and concise manner.

: Thank you very much for your positive reviews. The authors tried to revise this manuscript based on the reviewer’s suggestions as follows.

In the following there is a list of two minor issues that should be addressed before publication:

[Comment #1]: The test person collective is quite young. You should mention this potential bias.

: Thank you for your valuable comment. The authors fully agree with your opinion. So, this limitation has been added in the discussion section.

[Page 11, Lines 387-389: “This study has some limitations. First, this study was conducted only with young adults. Further study will recruit participants of various age groups and investigate the difference between the age groups.”]

[Comment #2]: I suggest adding error bars in Figs. 6-8 to visualize the standard deviation inside the data.

: Standard errors have been added in all figures. 

Round 2

Reviewer 1 Report

Dear authors,

Thank you for your corrections, the paper has been definitely improved.

However, I still have a major comment concerning the use of an average value of EMG into the Results section. Even to show an overall trend, it is not consistent to use an average value of EMG (for several muscles). If you still want to do it, you have to find somes references that supports this way of doing. If not, I really think it would be valuable to the paper to express EMG per muscle only.

Moreover, I don't understand the meaning of letters in caption of figures. Could you please detail the meaning of each letter?

Finally, standard deviations have been added into all figures. However, I am very sceptical about these values as they are all the same in all figures! Please review.

For these reasons, this paper still need to be reviewed from my opinion.

Best regards,

Author Response

# Reviewer 1

General comments:

The objective of this study is to assess the influence of wearing a chairless exoskeleton on muscle activity while performing a picking tasks at different heights. To this aim, the authors collected EMG data of 16 muscles during a simulated harvesting task at 6 different heights (40 to 140cm) with and without the wearable sitting support CEX. The introduction is well written however the objectives of the study have to be clarified, as the secondary objective is a consequence of the first one [Comment #7]. The methodology is well explained even if some details are missing (see below). The Results section tackled some questions although there have not been introduced, e.g. influence of gender and body parts. Thus, the definition of independent variables has to be clarified [Comments #2; #3, #11, #13, #17, #24]. The use of an average amount of EMG (across all muscles) is very questionable as it does not look biomechanically consistent [Comments #4, & #14]. To correspond to the aim of the study, main results should be dedicated to the interaction effects between use of the exoskeleton, working height, and muscle activity. Standard deviations are missing and the design of graphics can also be questioned [Comment #15]. In the Discussion section, the first paragraph about the influence of gender appears awkward and unnecessary [Comments #24]. Some outcomes in the Discussion are difficult to find in the Results section, like the fact that wearing the exoskeleton would decrease EMG activity by 30% [Comment #4, #26]. Consequently, for all these reasons, a major revision is recommended.

: Thank you for your valuable comments. The authors tried to reflect the reviewer's comments and revised this manuscript as follows.

 Specific comments:

Abstract

[Comment #1]: The distribution of “participants” has to be detailed in the abstract: number of women and men.

[Response #1]: Thank you for your valuable comment. As your request, the detailed information about participants has been added as follows. 

[Page 1, Lines 15-16: “Twenty healthy subjects (thirteen males and seven females) participated in this experiment.”]

[Comment #2]: The fact that “gender” is an independent variable is questionable as it does not correspond to the main objective of the current study. Removing this question might help clarify the paper.

[Response #2]: The gender section has been deleted as the reviewer’s comment to help clarify this study. 

[Comment #3]: The terms “body part” might be detailed as it refers to muscle activity on different anatomical areas.

[Response #3]: The authors agree with the reviewer’s comment. To clarify which muscle groups of the body were active, the authors replaced ‘body part’ with ‘muscle type’ in the manuscript.

[Page 1, Lines 16-19: “The independent variables were wearing of the exoskeleton (w/ CEX, w/o CEX), working height (6 levels: 40, 60, 80, 100, 120, 140 cm), and muscle type (8 levels:  upper trapezius (UT), erector spinae (ES), middle deltoid (MD), triceps brachii (TB), biceps brachii (BB), biceps femoris (BF), rectus femoris (RF), and tibialis anterior (TA))”.]

[Comment #4]: “the average muscle activity of participants wearing the CEX was approximately 30% lower than that of participants without the CEX”: this sentence is awkward as it gives the impression that two groups of participants did the experiment, please clarify. Moreover, the number “30% lower” is difficult to find into the Results section (see below).

[Response #4]: At the height of 40cm, W CEX (7.3% MVC) showed 36.0% lower EMG average amplitude than that of W/O (11.4% MVC) in Figure 7. The average amplitude of 60 and 80cm height also showed a similar pattern and showed 31.3% and 20.0% lower amplitude than W/O, respectively. To clarify the muscle activities in w/ and w/o the CEX for different working heights, the authors tried to present them based on each muscle group in this manuscript. So, it has been revised as follows.  

[Page 1, Lines 20-22: “When wearing the CEX, UT, ES, RF, and TA showed lower muscle activities at low working heights (40–80 cm) than not wearing the CEX, whereas those muscles showed higher muscle activities at high working heights (100-140 cm).”]

[Comment #5]: “… at a working height of 40-100cm might reduce the muscular load and discomfort of muscles and decrease the risk of related disorders.”: this part of the sentence has to be reviewed as it deals with “muscle activity” and not “muscle load”, and also “global discomfort” and not “discomfort of muscles”.

[Response #5]: Thank you for your comment. ‘Muscle load’ and ‘discomfort of muscles’ have been modified to ‘muscle activity’ and ‘discomfort of whole body’, respectively.

[Page 1, Lines 27-28: “~ 40-100cm might reduce the muscle activity and discomfort of whole body and decrease the risk of related disorders.”]

  1. Introduction

[Comment #6]: Page 2, l.58: “…to reduce exposure to prolonged stationary standing and sitting tasks”. The term “prolonged” is an important point here: wearable exoskeletons can be useful on a long-term task however the temporal aspect has not been studied in the current study, please justify.

[Response #6]: Thank you for your valuable comment. The authors very much agree with the reviewer's opinion. To verify the effectiveness of wearing EXO in the workplace, we think the study of the temporal aspect will be very important. First of all, the purpose of the current study was to research the effect of wearing EXO on muscle activity and subjective discomfort in a relatively short period of time, and we would like to conduct a prolonged task study later on.

The limitations of this study were added in the Discussion section.

[Page 11, Lines 388-390: “The second limitation is that this study did not consider the workload for a long-term period. In a further study, the tendency of workload over time would be considered by increasing the task time.”]

[Comment #7]: Page 3, l. 112: “… and (2) to provide guidelines for the appropriate working height for use of the lower-limb exoskeleton.”. This is not an objective but an application of the current study, please remove.

[Response #7]: The second purpose has been removed as requested. 

[Page 3, Lines 112-114: “Therefore, the purpose of this study was to evaluate the activities of upper and lower extremity muscles and subjective discomfort according to the heights of harvesting tasks and the use of a lower-limb exoskeleton (CEX).”]

  1. Methods

[Comment #8]: Page 3, l.122: “The elbow height…” Why has this distance been measured? It is not reused in the next sections and it does not support hypotheses of the study. Please clarify.

[Response #8]: The authors agree with the reviewer’s opinion. Thus, the elbow height has been deleted in Table 1 and section 2.1.

[Comment #9]: Page 3, l. 134: “…can be adjusted using Velcro straps to fit the worker’s body.”. Is there any anthropometrical requirement to wear the CEX (minimal and maximal stature, etc.)? Please detail.

[Response #9]: Thanks for your comment. The anthropometric requirement of the CEX has been added in the 2.2 section.

[Page 3, Lines 136-141: “The CEX has three different sizes (small, medium, and large), and users can choose the appropriate size according to their height and weight. The small size is suitable for users who are 160 to 175cm tall and weighing less than 80kg. The medium and large sizes are suitable for 170~185cm height (less than 100kg weight) and 180~195cm height (less than 120kg weight), respectively. In addition, the length of calf and hip and sitting angle are also adjustable over 3-5 levels to body sizes and working conditions.”]

[Comment #10]: Page 4, l. 162: “After donning the CEX, all participants performed sufficient pre-work to become familiar with the exoskeleton and the experimental task.”. How long was this familiarizing phase?

[Response #10]: In this study, 15 minutes of pre-work time was provided to all participants. It has been added in section 2.3.

[Page 5, Lines 167-168: “After donning the CEX, all participants performed sufficient pre-work for 15 minutes to become familiar with the exoskeleton and the experimental task.”]

[Comment #11]: Page 5, l. 179: the “gender” independent variable might be removed to improve the reading of the article. The terms “body parts” need to be replaced by “muscle activity” in the whole paper.

[Response #11]: Gender has been deleted in section 2.4, and authors thought that it is more appropriate to replace ‘body part’ with ‘muscle type’.

[Page 5, Lines 186-188: “The independent variables were the use of the exoskeleton (w/ CEX, w/o CEX), working height (6 levels: 40, 60, 80, 100, 120, 140 cm), and muscle type (8 levels: upper trapezius (UT), erector spinae (ES), middle deltoid (MD), triceps brachii (TB), biceps brachii (BB), biceps femoris (BF), rectus femoris (RF), and tibialis anterior (TA).”]

[Comment #12]: Page 6, l. 202: “… based on the traditional formula shown in equation (1).”. What is the reference related to this formula? The resting EMG is usually not removed and the maximal EMG (after a maximal voluntary contraction or the maximal value during the task) represents 100%. Current EMG is then expressed as a proportion of this maximum for normalization. Please justify.

[Response #12]: Normalization of EMG data is a method to expressing the task muscle activity as some percentage of the reference value. In this study, we used the maximum voluntary contraction (MVC) value as a maximum reference. Quite often, an EMG resting value is also used in order to quantify the resting level activity, and this could become a low-end reference (Soderverg, 1991; Serroussi and Pope, 1987), and the normalized EMG value more accurately reflects the muscle’s activity level required to perform the task (Mirka, 1991). Mirka (1991) defined this normalization method, and the equation is as follows [1]. Reference has been added in Method and Reference section.

[Page 6, Lines 209-210: “~ based on the traditional formula shown in equation (1) [35].”]

[Page 13, Line 491: “35. Mirka, G.A. The quantification of EMG normalization error, Ergon. 1991, 34(3), 343-352.”].

  1. Results

[Comment #13]: As previously mentioned, results concerning the effects of gender might be removed to clarify the paper.

[Response #13]: All results about gender have been removed. Thank you for your valuable comments.

[Comment #14]: An average EMG activity (for all muscles) is often used to express results in this study. However, this average amount is very questionable as it is not biomechanically consistent (none of the reference article in biomechanics used this average data). Indeed, each muscle has its own characteristics so averaging EMG through different muscles would remove this specificity. An EMG-based analysis should be performed per muscle only. The Results section should be thus reviewed consequently.

[Response #14]: Thank you for your valuable comments. In Figure 5 (left) and 7 in the Result section, the average value of EMGs was inevitably used to examine the overall trend of muscle activities associated with working height and the interaction effect between working height and use of exoskeleton. Although the authors agree with the reviewer’s comment, these figures would be a useful way to understand the overall trend. However, as the comment of the reviewer (the authors also understand that the analysis of each muscle is important), the results of each muscle are presented through Figures 8, 10, and 11. 

As the reviewer’s comment, some of the contents where the average value was used have been removed, and muscle-specific analysis has been added in the Abstract and Discussion section.

[Page 1, Lines 20-22: “When wearing the CEX, UT, ES, RF, and TA showed lower muscle activities at low working heights (40–80 cm) than not wearing the CEX, whereas those muscles showed higher muscle activities at high working heights (100-140 cm).”]

[Page 10, Lines 335-337: “In lower working height (40-80cm), which requires a high degree of knee and back flexion, most muscles (UT, ES, RF, and TA) showed lower muscle activity when wearing the CEX than that of when not wearing the CEX.”]

[Page 10, Lines 341-345: “UT muscle showed significantly reduced muscle activities when wearing the CEX by 35.93% (w/o CEX: 16.7% and w/ CEX: 10.7%) and 50.34% (w/o CEX: 14.7% and w/ CEX: 7.3%) at working height of 40 and 60cm, respectively. Similarly, ES muscle also showed reduced muscle activity (5.83-29.48%) when wearing the CEX at a working height of 40-80cm (w/o CEX: 12.0-17.3% and w/ CEX: 10.4-12.2%)”]

[Pages 10, Lines 354-356: “In particular, at working heights of 100-140cm, the UT muscle showed about 2.3-3.0 times greater muscle activity (1.9-22.6%) when working with CEX than when working without CEX (2.2-6.4%).”]

[Comment #15]: Standard deviations (SD) are missing in all results, both in the text and graphics, please add.

[Response #15]: Based on the reviewer 1, 2, and 3, standard errors have been added in all figures.  

[Comment #16]: Figure 5: what are letters (A, B, C, D, AB, BC) referring to? Please detail in the caption.

[Response #16]: Thank you for your comment. The caption has been added in Figure 5 as follow.

[Page 7, Lines 238-239: “Different letters indicate significant statistical differences (p< 0.05, Tukey’s test).”]

[Comment #17]: Figure 6: please remove as it refers to the question of gender.

[Response #17]: Figure 6 and related contents have been removed based on the reviewer's request.

[Comment #18]: Figure 7: what are letters (a, b, A, AB, etc.) referring to? Please detail in the caption.

[Response #18]: The caption has been added in Figure 7.  

[Page 7, Lines 258-259: “Asterisk (*) shows a significant difference between W/O and W (p<0.05, Paired t-test). Different letters indicate significant statistical differences (p<0.05, Tukey’s test).”]

[Comment #19]: Figure 8: why using lines between muscle EMG data? There is no rationale for that, please use histograms. SD are missing and significance should be integrated into the figure.

[Response #19]: As the reviewer’s comments, Figure 8 has been revised.

[Page 8, Lines 270-272: “Asterisk (*) shows a significant difference between W/O and W (p<0.05, Paired t-test). Different letters indicate significant statistical differences (p<0.05, Tukey’s test).”]

Figure 7. Interaction effect between use of the exoskeleton and muscle type. Asterisk (*) shows a significant difference between W/O and W (p< 0.05, Paired t-test). Different letters indicate significant statistical differences (p< 0.05, Tukey’s test).

{Figure 8 has been changed to Figure 7 due to the deletion of Figure 6}

[Comment #20]: Figure 9: what are letters referring to? What is the aim of this figure? Showing that EMG activity is different with heights, wearing or not an exoskeleton? This is obvious and does not correspond to any question from the literature. Thus, I would propose to remove the interaction between working heights and muscle activity from the analysis as it undermines the main message of the paper.

[Response #20]: Thank you for your valuable comment. The authors agreed with the reviewer’s opinion. Figure 9 and contents have been removed in this manuscript.

[Comment #21]: Figure 10 and 11: explanation about letters, SD and significance are necessary.

[Response #21]: Figures 10 and 11 have been revised. The results of paired t-test (W/O vs. W) have also been added. The bar graphs have been replaced with line graphs to improve legibility.

 [Page 8, Lines 287-289 & Page 9, Lines 298-300: “Asterisk (*) shows a significant difference between W/O and W (p< 0.05, Paired t-test). Different letters indicate significant statistical differences (p< 0.05, Tukey’s test).”]

Figure 8. Interaction effects among wearing exoskeleton, working height, and muscle types (UT and ES). Asterisk (*) shows a significant difference between W/O and W (p< 0.05, Paired t-test). Different letters indicate significant statistical differences (p< 0.05, Tukey’s test).

{Figure 10 has been changed to Figure 8 due to the deletion of Figures 6 and 9}

Figure 9. Interaction effects among wearing exoskeleton, working height, and muscle type (BF, RF, and TA). Asterisk (*) shows a significant difference between W/O and W (p< 0.05, Paired t-test). Different letters indicate significant statistical differences (p< 0.05, Tukey’s test).

{Figure 11 has been changed to Figure 9 due to the deletion of Figures 6 and 9}

[Comment #22]: Page 10, l.313: “The interaction effect between working height and use…” it is necessary to precise that it concerns subjective ratings in this sentence.

[Response #22]: The sentence has been revised as requested.

[Page 9, Line 308-3091: “The interaction effect between working height and use of the exoskeleton on the subjective discomfort rating was also statistically significant (p= 0.001).”]

[Comment #23]: Figure 12: the two graphs need to be described in the caption. Explanation about letters, SD and significance are necessary.

[Response #23]: Figures 12 has been revised.

[Page 9, Lines 317-3181: “Asterisk (*) shows a significant difference between W/O and W (p< 0.05, Paired t-test). Different letters indicate significant statistical differences (p< 0.05, Tukey’s test).”]

Figure 10. Interaction effect between exoskeleton use and working height. Asterisk (*) show significant difference between W/O and W (p<0.05, Paired t-test). Different letters indicate significant statistical differences (p<0.05, Tukey’s test).

{Figure 12 has been changed to Figure 10 due to the deletion of Figures 6 and 9}

  1. Discussion

[Comment #24]: The first paragraph about the influence of gender is awkward and should be removed.

[Response #24]: All contents related to the gender effect have been removed.

[Comment #25]: Page 11, l.336: “In particular, at lower working heights, …”. This sentence is too long and should be segmented.

[Response #25]: Thank you for your comment. That sentence has been condensed.

[Page 10, Lines 367-368: “In particular, at lower working heights (40-80 cm), the overall discomfort rating of using the CEX was significantly more positive than that of not using the CEX.”]

[Comment #26]: Page 11, l.346: “… was approximately 30% lower…” I don’t find where this number comes from. Is based on average EMG? If yes, it has to be reviewed. This kind of sentence might take place if all muscles present a decrease of 30% of activity in all conditions with the exoskeleton, which is not the case.

[Response #26]: Thank you for your comments. As recommendations, the average data of muscles have been replaced with muscle-specific results.  

[Page 10, Lines 336-338: “In lower working height (40-80cm), which requires a high degree of knee and back flexion, most muscles (UT, ES, RF, and TA) showed lower muscle activity when wearing the CEX than that of when not wearing the CEX.”]

[Page 10, Lines 342-346: “UT muscle showed significantly reduced muscle activities when wearing the CEX by 35.93% (w/o CEX: 16.7% and w/ CEX: 10.7%) and 50.34% (w/o CEX: 14.7% and w/ CEX: 7.3%) at working height of 40 and 60cm, respectively. Similarly, ES muscle also showed reduced muscle activity (5.83-29.48%) when wearing the CEX at a working height of 40-80cm (w/o CEX: 12.0-17.3% and w/ CEX: 10.4-12.2%)”]

[Pages 10, Lines 355-357: “In particular, at working heights of 100-140cm, the UT muscle showed about 2.3-3.0 times greater muscle activity (1.9-22.6%) when working with CEX than when working without CEX (2.2-6.4%).”]

[Comment #27]: Page 11, l.360: “… also showed slightly…” the word “slightly” needs to be removed as it is not scientifically consistent, please be more accurate (significant or not?).

[Response #27]: Thank you for your comments. As the reviewer’s comment, this sentence has been revised more clearly. For clarity of this study, the result of paired t-test between W and W/O CEX has been added in Figures 10 & 11.

[Page 10, Lines 357-359: “TA muscles also significantly increased muscle activity when wearing the CEX than when not wearing the device at higher working heights (100, 120, 140cm).”]

  1. Miscellaneous

[Comment #28]: Page 3, l. 100: “…(Chirless chair, LegX)…” the word “chirless” has to be replaced by the word “chairless”

[Response #28]: Typo has been revised.

Reviewer 2 Report

no comments now

Author Response

# Reviewer 2

This study used a wearable passive exoskeleton to compare EMG during various work tasks, and the results are interesting. However, there are some concerns to be clarified.   

: Thank you for your valuable comments. The authors tried to reflect the reviewer's comments and revised this manuscript as follows.

[Comment #1]: Line 135, could the length or height of the exoskeleton be adjusted? If not, the protocol had a problem for different individuals.

[Response #1]: Thank you for your valuable comments. The CEX has three types of sizes (S, M, and L), and users can choose the appropriate size according to their height and weight. In addition, the length of calf and hip and sitting angle of CEX can be adjusted in 3-5 steps. It has the advantage of being able to adjustable according to the user’s body. More information about the CEX has been described in this manuscript.

[Page 3, Lines 136-141: “The CEX has three different sizes (small, medium, and large), and users can choose the appropriate size according to their height and weight. The small size is suitable for users who are 160 to 175cm tall and weighing less than 80kg. The medium and large sizes are suitable for 170~185cm height (less than 100kg weight) and 180~195cm height (less than 120kg weight), respectively. In addition, the length of calf and hip and sitting angle are also adjustable over 3-5 levels to body sizes and working conditions.”]

[Comment #2]: Line 176, what specific functions were selected in Spss?

[Response #2]: Analysis of variance (ANOVA) and paired t-test were used in SPSS. Related information has been added in the method section.

[Page 5, Lines 182-183: “Analysis of variance (ANOVA) and paired t-test were conducted to determine the effect of the CEX and working height on EMG activities and subjective rating.”]

[Comment #3]: Line 204, how to determine or collect rest EMG signals, e.g. standing or siting?

[Response #3]: Thank you for your comments. To measure the resting EMG of the ES muscle, all participants were asked to lie on a pad in a prone position and asked to be relaxed for 5 seconds, and it was repeated two times. The resting EMGs of other muscles were also measured in sitting posture for two trials and used the average value of two trials in this study.

[Comment #4]: Line 207 and 208, what is RMS and how to calculate it? What is ECG filters? Please give detailed introduction.

[Response #4]: Thank you for your comments. First, the RMS represents the squared root of the average poser of the EMG signals for a given time. It is the recommended quantitative index of muscle activity for surface EMG signals (Basmajian and De Lura, 1985). Related information has been added in the method and reference sections.

Secondly, ECG (electrocardiogram) spikes can interfere with the EMG recording of muscles, especially when performed on the upper trunk & shoulder movements. Therefore, an ECG reduction filter has been widely used to eliminate ECG signals. Detailed information has been added in the method section.

[Page 6, Line 216: “~ then expressed as the root mean squared (RMS, 50 ms) signal [36].”]

[Page 6, Lines 217-219: “Electrocardiogram (EGC) spikes can interfere with the EMG signals of muscles, especially when performed on the upper trunk and shoulder activities. Thus, an ECG reduction filter was applied to eliminate EGC signals in this study.”]

[Comment #5]: Figure 5, where are SD or SER?

[Response #5]: Standard errors have been added in all figures.  

[Comment #6]: In Figure 6 and other figures, what meanings of superscripts a,b,c,d..or A,B,C,D…etc? Also in Figure 6, with or without the wearable exoskeletal device?

[Response #6]

: Thank you for your valuable comments. Figure 6 has been deleted as reviewer 1’s opinion. However, as per your comment, the caption has been added to all figures. For example,

[Page 7, Lines 259-259: “Asterisk (*) shows a significant difference between W/O and W (p< 0.05, Paired t-test). Different letters indicate significant statistical differences (p< 0.05, Tukey’s test).”]